# Intensified Circulation of Echovirus 11 after the COVID-19 Pandemic in Poland: Detection of a Highly Pathogenic Virus Variant

**DOI:** 10.3390/v16071011

**Published:** 2024-06-24

**Authors:** Beata Gad, Paulina Kłosiewicz, Kinga Oleksiak, Arleta Krzysztoszek, Kacper Toczyłowski, Artur Sulik, Tobiasz Wieczorek, Magdalena Wieczorek

**Affiliations:** 1Department of Virology, National Institute of Public Health NIH—National Research Institute, Chocimska 24, 00-791 Warsaw, Poland; bgad@pzh.gov.pl (B.G.); pklosiewicz@pzh.gov.pl (P.K.); koleksiak@pzh.gov.pl (K.O.); akrzysztoszek@pzh.gov.pl (A.K.); 2Department of Pediatric Infectious Diseases, Medical University of Bialystok, Waszyngtona 17, 15-274 Bialystok, Poland; kacper.toczylowski@umb.edu.pl (K.T.); artur.sulik@umb.edu.pl (A.S.); 3Faculty of Civil Engineering and Geodesy, Military University of Technology, Gen. S. Kaliskiego 2, 00-908 Warsaw, Poland; tobiasz.wieczorek@wat.edu.pl

**Keywords:** echovirus 11, phylogenetic analysis, environmental surveillance, enterovirus, epidemiology, aseptic meningitis, multisystem disease

## Abstract

After the first phase of the COVID-19 pandemic in Europe, a new highly pathogenic variant of echovirus 11 (E11) was detected. The aim of this study was to analyze the genetic diversity of Polish E11 environmental and clinical strains circulating between 2017 and 2023 as well as compare them with E11 strains isolated from severe neonatal sepsis cases reported in Europe between 2022 and 2023. Additionally, the study explores the effectiveness of environmental monitoring in tracking the spread of new variants. For this purpose, the complete sequences of the VP1 capsid protein gene were determined for 266 E11 strains isolated in Poland from 2017 to 2023, and phylogenetic analysis was performed. In the years 2017–2023, a significant increase in the detection of E11 strains was observed in both environmental and clinical samples in Poland. The Polish E11 strains represented three different genotypes, C3, D5 and E, and were characterized by a high diversity. In Poland, the intensive circulation of the new variant E11, responsible for severe neonatal infections with a high mortality in Europe, was detected in the years 2022–2023. This investigation demonstrates the important role of environmental surveillance in the tracking of enteroviruses circulation, especially in settings with limited clinical surveillance.

## 1. Introduction

Echovirus 11 (E11) is one of the most common enteroviruses detected in both clinical and environmental samples worldwide [1,2,3,4,5,6]. It belongs to the B species of the Enterovirus genus of the Picornaviridae family. Though E11 is known mainly for mild or asymptomatic infections, it is responsible for severe syndromes, characterized by high rates of morbidity and mortality [7]. The wide range of diseases encompass aseptic meningitis, encephalitis, acute flaccid paralysis (AFP), upper respiratory tract infections, hand, foot and mouth disease (HFMD), acute gastroenteritis, myocarditis and uveitis [8,9,10,11,12]. In particular, this virus can cause severe systemic infection in newborns, including acute hepatitis with coagulopathy, which is one of the most fatal complications of severe neonatal enterovirus infections [13,14]. As with other enteroviruses, E11 has the ability to cross the placenta and infect the fetus, leading to heart disease and fetal death [15,16,17]. People with primary immunodeficiency are also at risk of chronic infection caused by E11 [18]. This virus has an epidemic pattern of circulation and has been frequently identified as the causal agent of outbreaks that occur irregularly and often last for several years [1]. Similarly, E11 has been associated with nosocomial infections and numerous outbreaks in neonatal nurseries [19,20,21].

The infection outbreaks and silent circulation within the population can be tracked through environmental surveillance. Testing sewage samples allows for the detection of a wider spectrum of virus variants circulating in the population compared to those isolated from patients. Specifically, raw sewage contains viruses shed from both asymptomatic and symptomatic individuals. Moreover, it was proven that the pathogens detected in sewage were highly correlated with clinical cases and disease outbreaks [5,22,23].

The World Health Organization (WHO) and the European Center for Disease Prevention and Control (ECDC) have reported an increasing number of severe neonatal infections associated with E11 in 2022–2023 [24,25,26,27,28]. E11 was found in neonatal patients with severe sepsis, complicated by liver failure and neurological or myocardial involvement. The lethality rate of severe neonatal E11 infection was very high. Consequently, a new variant of E11 was reported from France, Croatia, Italy, Spain, Sweden and the United Kingdom of Great Britain and Northern Ireland [29,30].

Mutation and recombination play crucial roles in the formation of new genetic variants of E11 with novel biological properties associated with changes in tissue tropism, the evasion of host immunity, increased virulence and pathogenesis and the emergence of disease outbreaks. VP1 is the most immunodominant structural protein of E11, and antigenic variants are selected in the presence of antibodies to it. Molecular methods based on the amplification and sequencing of the VP1 coding region (1D) are used for the molecular characterization of strains and epidemiological studies. E11 strains circulating worldwide are classified into six genotypes (A–F), and genotypes A, C, and D are divided into A1–5, C1–4 and D1–5 [31,32].

The aim of this study was to examine the range of genetic variation within the Polish environmental and clinical E11 strains circulating between 2017 and 2023 and to compare them with E11 strains isolated from severe cases of neonatal sepsis reported in 2022 and 2023 in Europe. The phylogenetic relationship with other worldwide circulating strains was also analyzed. For this purpose, the complete sequences of the VP1 capsid protein gene were determined for E11 strains isolated in Poland from 2017 to 2023. Furthermore, this study addresses the utility of environmental surveillance for tracking the circulation of new variants of viruses.

## 2. Materials and Methods

### 2.1. Viruses

In total, 266 isolates of E11 were obtained from the collection of the National Institute of Public Health, NIH-NRI in Poland. Two hundred fifty-five of them were isolated from sewage samples collected in 2017–2023, and eleven were isolated from clinical cases (aseptic meningitis, neonatal sepsis) in 2018–2019 and 2022–2023. The environmental strains were isolated from sewage samples collected in five Polish cities at different time intervals: Warsaw (2017–2023, one to eight collection sites), Lublin (2022–2023, one to eight collection sites), Rzeszow (2022–2023, one collection site), Krakow (2023, one collection site), and Gdansk (2023, one collection site). The frequency of sampling varied during the study period, with collections occurring once a week from July 2017 to March 2020, once a quarter in 2021, and twice a month in 2022–2023. Sewage samples were concentrated according to the protocol described earlier [33] and inoculated onto two cell lines, L20B and RD.

The isolates of E11 were propagated in RD (rhabdomyosarcoma) cell cultures according to the instructions described in Polio Laboratory Manual, 4th edition, 2004, World Health Organization [34]. RD cells were maintained in MEM (Minimum Essential Medium) supplemented with 10% FBS (Fetal Bovine Serum).

### 2.2. RNA Isolation and RT-PCR

Viral RNA was extracted from the cell culture supernatant using a QIAamp Viral RNA Mini Kit (Qiagen, Hilden, Germany), following the manufacturer’s instructions. The complete VP1 coding region was amplified by reverse transcription PCR using Superscript III (Invitrogen, Carlsbad, CA, USA), specific primers and PCR cycling times and temperature, as previously described [35]. The amplified products were analyzed in 1.5% agarose gels, GelRed-stained and examined under a UV DNA transilluminator.

### 2.3. Sequencing and Sequence Analysis

The PCR products were processed in a cycle sequencing reaction with BigDye 3.1 (Applied Biosystems, Foster City, CA, USA) according to the manufacturer’s protocol. The product of the sequencing reaction was run in an automated genetic analyzer (Applied Biosystems, Foster City, CA, USA, model 3730). The resulting sequences were manually edited using BioEdit version 7.1.9 and examined in terms of the closest homologue sequences using BLAST software, version **BLAST+ 2.15.0** (http://www.ncbi.nlm.nih.gov/BLAST/, accessed on 1 March 2024). Phylogenetic and molecular evolutionary analyses were conducted using MEGA version X [36]. The sequences of isolated strains were aligned with the reference strains and a selection of sequences from worldwide strains. In total, 344 VP1 sequences were included in the phylogenetic analysis, including the 266 sequences generated in this study. A phylogenetic tree was computed using the neighbor-joining method with a bootstrap of 1000 replicates. All sequences obtained in this study were submitted to GenBank and were assigned accession numbers from PP534556 to PP534821.

In total, 150 VP1 sequences were included in the haplotype analysis, along with the 132 Polish sequences from 2018 to 2023, 17 European sequences from 2022 to 2023 and 1 sequence from the USA (2022). Median-joining network analysis was performed using PopART (Population Analysis with Reticulate Trees) software version 1.7 based on multisequence aligned haplotype data and traits data representing the geographical distribution [37].

## 3. Results

### 3.1. E11 Detection during 2017–2023

In total, 1197 NPEV isolates were collected at the National Institute of Public Health, NIH-NRI (NIPH NIH-NRI) in Poland, including 266 E11 isolates (266/1197; 22.22%), during 2017–2023. Most of the E11 came from environmental samples. The environmental strains of E11 accounted for 30.36% of the total NPEV isolated from sewage samples (255/840); the clinical strains of E11 represented only 3.08% of the total NPEV isolated from clinical samples (11/357). E11 was detected during each of the reported years. The number of isolated E11 isolates varied in particular years (Figure 1). The lowest number of E11 isolates, both clinical and environmental, was observed in 2017, and the highest was observed in 2023. The number of E11 isolates increased significantly from 2017 (*n* = 7) to 2023 (*n* = 124), representing 5.74% and 43.20% of the total number of isolates detected for the respective years. NPEV isolations, including E11, during the COVID-19 pandemic, especially during 2020 and 2021, were considerably lower than those after and before the pandemic. E11 was not detected in clinical samples during 2020 and 2021 at NIPH NIH-NRI, suggesting that a reduced transmission of E11 occurred during the intense phases of non-pharmaceutical interventions and social mitigations of COVID-19 in Poland.

### 3.2. Geographic and Temporal Distribution of E11 Strains

The environmental strains were isolated from sewage samples collected in five Polish cities, in different time intervals (Warsaw 2017–2023, Lublin 2022–2023, Rzeszow 2022–2023, Krakow 2023, Gdansk 2023), while the clinical isolates came from four different regions from 2018 to 2019 and from 2022 to 2023 (Masovian Voivodeship, Lesser Poland Voivodeship, Kuyavian-Pomeranian Voivodeship, Podlaskie Voivodeship) (Figure 2). In the cities where environmental samples were collected, an increasing trend of E11 detection was observed in subsequent years of the study. In 2023, environmental E11 isolates represented more than 40%, while in 2017–2022, they represented less than 40% of the total number of NPEV detected in a given year at each collection site. In the individual sampling cities, E11 constituted 0 (Lublin, 2022) to 82.61% (Gdansk, 2023) of the total NPEV obtained. The monthly distribution of environmental E11 is shown in Figure 3. E11 isolates were frequently isolated from environmental samples in 2023 and obtained throughout the year. In the years 2017–2023, E11 was most often isolated in the summer and autumn months (from July to December), peaking in November (Figure 3).

### 3.3. Clinical Characteristics of E11 Cases

Between 2017 and 2023, 11 E11 clinical isolates were identified in NIPH NIH-NRI (Table 1). Two isolates were obtained in 2018–2019, and the remaining nine were obtained in 2022–2023. Of all the clinical isolates from 2023, more than half were associated with infection in infants and toddlers. The isolates were generally obtained from patients with symptoms of aseptic meningitis. Most of the mentioned cases had a mild course, except one case of systemic infection with hepatitis. Cerebrospinal fluid (CSF) examination showed pleocytosis with white blood cells (WBC) ranging from 7 to 263 cells/µL in Polish E11 infection cases. Female patients (6/11, 54.55%) and infants <3 months of age (6/11, 54.55%) represented more than half of the reported cases.

### 3.4. Phylogenetic Analysis of E11 Strains

In this analysis, a total of 266 E11 isolates derived from clinical and environmental samples, collected between 2017 and 2023 in Poland, were entered. To evaluate the phylogenetic relationships among circulating E11 in Poland, full-length VP1 sequences (876 bp) of 266 isolated strains were investigated. The presented phylogenetic tree (Figure 4) was constructed based on 282 sequences, with 204 generated in this study. Sequences that were identical or highly similar were not included. The study isolates were grouped into three different genotypes, C3, D5 and E (Figure 4), according to the nomenclature proposed by Oberste et al. and Li et al. [31,32]. All clinical isolates (*n* = 11) and most environmental isolates (*n* = 244) were clustered in genotype D5. Nine environmental isolates from 2019 and 2021–2023 were placed in genotype E together with isolates from Eurasia and Africa (1992–2012). Two environmental isolates from 2017 and 2023 were placed in genotype C3 along with isolates from China (2014–2015), Russia (2011) and Algeria (2008). Sequences of clinical isolates from 2022 to 2023 were grouped together with those from France and Italy from 2022 to 2023.

In general, nucleotide sequence divergence in pairwise comparisons between Polish E11 isolates ranged from 0.0% to 25.7% (0.0–12.7% aa divergence) (Table 2) and depended on the year of detection. In 2020, it varied from 0.0% to 8.3% (0.0–2.7% aa divergence) and in 2023, it varied from 0.0% to 25.3% (0.0–12.3% aa divergence) when the three genotypes were detected. The viruses of E showed the highest nucleotide divergence (0.0–15.1%), and D5 showed the smallest (0–9.9%). Furthermore, the nucleotide diversity of environmental isolates was significantly higher than that of clinical E11. Polish E11 strains showed 75.6–81.2% nucleotide and 87.3–94.2% amino acid similarity with the prototype strain—Gregory.

The complete VP1 coding sequence of E11 consisted of 292 aa, and 69.18% of aa sites (202/292) were conserved between the Polish strains. A significant number of aa substitutions were observed; 90 sites of 292 aa residues had been changed between isolates. In most polymorphic sites, amino acid substitutions were not associated with clustering (49/90, 54.44%). Forty-one sites showed specific aa conservation for genotypes, and most of them were located in the amino and carboxyterminal region of the VP1 protein or in loops (Table 3). The conservation of aa signatures specific to these genotypes was incomplete, showing a return to the original state at locations 23, 84 and 292 in the C3 genotype and at locations 84, 245, 271 and 280 in the E genotype.

The generation of haplotype networks is a widely used methodology for analyzing and visualizing the relationships between sequences from different geographic destinations. A cluster of severe neonatal cases associated with a new variant of E-11, which contains, among others, sequences from France (2022–2023), Italy (2023) and Poland (2017–2023), was analyzed using PopArt software version 1.7. A total of 150 D5 genotype sequences were subjected to the median-joining haplotype network analysis. The analysis showed a close relationship of Polish environmental and clinical isolates with clinical cases from France, Italy and Great Britain, associated with severe cases of neonatal infections in Europe. The haplotype analysis also revealed that most of the strains from France were separated by as few as 0–6 nucleotide substitutions in the VP1 gene from Polish environmental or clinical strains. The French clinical isolate from 2022 shares an identical VP1 gene sequence with the Polish environmental isolate from the same year. Additionally, the Polish clinical strain (PL14/4786/2022, PP534560), isolated from a severe case of hepatitis and multisystemic disease, clusters with clinical strains from France and Great Britain associated with multiorgan failure, neonatal sepsis and sudden infant death, as well as with Polish environmental strains from Gdansk and Warsaw (depicted by a grey triangle in Figure 5).

## 4. Discussion

In this study, we conducted a genetic comparison of E11 strains isolated from clinical samples with environmental E11 strains collected in Poland from 2017 to 2023. This research was prompted by the emergence of severe neonatal E11 infections with high mortality rates reported by the World Health Organization (WHO) and the European Center for Disease Prevention and Control (ECDC) in Europe during 2022–2023. Between 2022 and July 2023, WHO and ECDC reported 21 cases of severe sepsis caused by E11 infection, which were complicated by liver failure and neurological or myocardial involvement. These cases occurred in countries including France, Croatia, Italy, Spain, Sweden and the United Kingdom of Great Britain and Northern Ireland [24,25,26,27,28,29,30] and were related to the emergence of a new variant of E11. Our analysis revealed the intensive circulation of the same variant in Poland between 2022 and 2023. Both Polish environmental and clinical isolates were closely related to strains from France, Italy and Great Britain. The virus collection from the Department of Virology, NIPH NHI-NRI, comprised nine clinical isolates of E11 from 2022 to 2023, with only one isolate associated with a severe case of sepsis accompanied by hepatitis. Over half of these E11 infections (58%) were reported in infants under the age of three months old. This reiterates the previous observations that E11 infections were most commonly notified in children younger than 3 months old [2]. Furthermore, the number of E11 clinical isolates was significantly higher compared to those of the years before the COVID-19 pandemic (2017–2019; *n* = 2). Numerous authors are addressing the spread of viruses in the midst of and following the COVID-19 pandemic. The reduction in virus circulation during the COVID-19 pandemic is believed to have affected the severity of new infections due to the prolonged absence of ongoing natural exposure to viruses. Lower levels of population immunity, especially among younger children, may result in a greater prevalence of disease and potentially more severe infection when the virus circulation resumes. Research conducted in the USA showed a significant disruption in seasonal EV detections during the early phases of the COVID-19 pandemic and a rapid increase in detections during the summer and fall of 2022 [38]. Similar disturbances in EV circulation have been observed in the Netherlands during and after lockdown [39].

Generally, E11 is one of the most commonly reported kinds of NPEV in the world. Between 1970 and 2005, in the USA, E11 represented 11.4% of all reported isolates from clinical samples [1]. A similar situation occurred in China, where E11 accounted for 11.2% of all the detected NPEV from cerebrospinal fluid samples during 2018–2019 [3,40]. In Europe, in turn, E11 was among the ten most frequently reported types, representing 4% of all typed EV positive samples between 2015 and 2017 [2]. However, due to the fact that most enterovirus infections are generally asymptomatic or mildly symptomatic, it makes it difficult to trace infections through clinical surveillance. Therefore, environmental surveillance is a more effective method of detecting viruses that circulate silently.

Environmental studies reveal more clearly that E11 is among the most frequently detected enteroviruses across the world. Chinese environmental reports indicated the high frequency isolation of E11 from sewage samples in different time periods (26.4%—242/916, 2009–2012 [4], 26.3%—336/1279, 2013–2021 [5]). E11 was also among the most frequently isolated enteroviruses in environmental surveillance, for example, in the USA during 1994–2002 [41], in Russia during 2004–2017 [42], in India during 2007–2009 [43], in Italy during 2009–2015 [44] and in Japan during 2013–2016 and 2019–2021 [45,46]. Our study revealed a significantly higher presence of E11 in the environmental samples in 2022–2023. E11 represented more than 60% of all environmental NPEV strains collected in 2023 and was detected throughout the year, indicating a change in the seasonal pattern of circulation. In our previous environmental studies conducted in 2011, E11 was also the most prevalent enterovirus (26%, 28/107); however, by 2023, it had surpassed its previous level of frequency [6]. The research carried out in Sicily also showed a sudden increase in the detection of E11 in 2023 [47], which may indicate an increase in the circulation of E11 throughout Europe.

Epidemics and outbreaks caused by E11 are often accompanied by severe cases with high mortality. The epidemic in Hungary in 1989 is an example, where 13 fatal cases of hemorrhagic syndromes were recorded among 386 children suffering from E11-associated disease (fever, vomiting, diarrhea) [48]. The National Enterovirus Surveillance System in the United States showed that E11 was the most commonly isolated EV from newborns during 1983–2003, representing 14% of neonatal EV infections, 19% of which were fatal [49]. This information should be related to the fact that in a similar time period (1970–2005), E11 was the second most frequently detected enterovirus in the USA [1]. The intense circulation of E11 in Poland and throughout Europe during 2022–2023 may explain the occurrence of severe cases in newborns in many countries in Europe. The lack of specific maternal enterovirus antibodies, especially E11, can lead to an increased number of severe cases and a higher fatality rate of neonatal enterovirus infections. Some studies suggest that the level of neutralizing antibodies to E11 in cord sera is lower compared to that of other serotypes and also lower than in maternal sera, which may determine the severity of E11 infection in newborns [17,50].

In this study, phylogenetic analysis showed that E11 isolates from 2017 to 2023 belong to three genotypes, C3, D5 and E. Polish isolates of genotypes C3 and E were found only in sewage, while D5 isolates were detected in both sewage and human samples. The D5 genotype was predominant in Poland during 2017–2023 and was consistently detected over a 7-year research period. The close relationship between the Polish D5 strains and European clinical strains suggests sustained circulation in the region during the study period. Previous studies reported the detection of D5 on four different continents, indicating the cosmopolitan nature of this genotype [32]. Severe cases of multisystemic infection in newborns in Europe during 2022–2023 were linked to a new variant of this genotype [29]. The emergence of a new enterovirus variant may stem from either the introduction of a new virus from a geographically distinct area or the genetic evolution of an endemic virus population. Phylogenetic analysis suggests that the new E11 variant emerged from an endemic circulating population through evolutionary processes.

Our analysis showed a high nucleotide diversity (up to 25.7%) between Polish E11 strains and a high number of amino acid substitutions in VP1. Phylogenetic data from previous studies have shown a greater diversity range of E11, with variations of up to 27.6% [32]. The high variety and low genetic stability reported for E11 may facilitate the emergence of new variants. It is worth noting that such high variation was revealed in our study through the analysis of environmental strains isolated from sewage samples. Notably, environmental surveillance allowed for the detection of two genotypes (C3 and E) in Poland. The fact that those genotypes were not isolated from clinical cases suggests silent circulation in the Polish population. It could be stated that only a small amount of E11 circulating causes detectable symptomatic infections. Interestingly, the Chinese researchers also described the circulation of several E11 genotypes at the same time, albeit causing only sporadic cases [31]. The study from Russia showed that certain genotypes of E11 were more prevalent in sewage, compared to other genotypes like D5, which were isolated from both sewage and human samples [51].

There are certain limitations to the study. The primary issue is the insufficient data regarding the number of E11 cases in Poland following the initial phase of the COVID-19 outbreak. Due to the absence of clinical surveillance for NPEV in Poland, sporadic cases, including potentially severe ones, of E11 may have gone unnoticed. Environmental surveillance was not conducted with the same frequency throughout the study period, especially during 2020–2021, which was affected by the COVID-19 pandemic. This limitation could have led to an underestimated number of enterovirus isolates obtained. Moreover, environmental surveillance was primarily conducted in eastern Poland, potentially limiting the representativeness of the analyzed isolates for the entire country. Furthermore, the detection method, which involves isolation in cell cultures, can preferentially detect certain types of enteroviruses—for example, E11, which can multiply well in RD cells. Multiple studies confirmed that RD cells support more replication enteroviruses of species B and polioviruses compared to other species [52]. Other studies have reported that coxsackie B viruses are more frequently isolated from Hep-2 cells than from RD cells [40]. Therefore, it is possible that our research did not fully reflect the actual proportions of enteroviruses in the sewage. Moreover, the frequency of isolation of different serotypes depends on individual characteristics, such as different rates of multiplication or varying stability in sewage. This could result in a higher frequency of isolating certain enteroviruses, such as E11, compared to others. It should also be mentioned that the identification of enterovirus types has some limitations due to the methods and primers used [53]. Additionally, it is important to note that the phylogenetic analysis focused on a fragment encoding one viral protein rather than the entire genome, leaving it unclear whether recombination has occurred.

## 5. Conclusions

The research carried out on environmental samples indicated an intensive circulation of E11 in Poland in 2022–2023. Taking into account the close genetic relationship between isolates recovered in Poland and other European countries, we can conclude that the increase in infection events in Europe caused by E11, including the presence of severe cases, was related to the intensive circulation of the D5 genotype in the European population. Taking into account the scale of detection of E11 in environmental samples, the serious cases of infections reported in Europe probably constituted a small percentage of asymptomatic and mildly symptomatic infections.

This study contributes to a better understanding of the circulation patterns of E11 in humans and their possible implication in severe cases of multisystemic disease in neonates. Furthermore, the research conducted indicates the need for constant clinical enteroviral surveillance supplemented with environmental surveillance, which allows for a more complete assessment of the epidemiological situation.

## Figures and Tables

**Figure 1 viruses-16-01011-f001:**
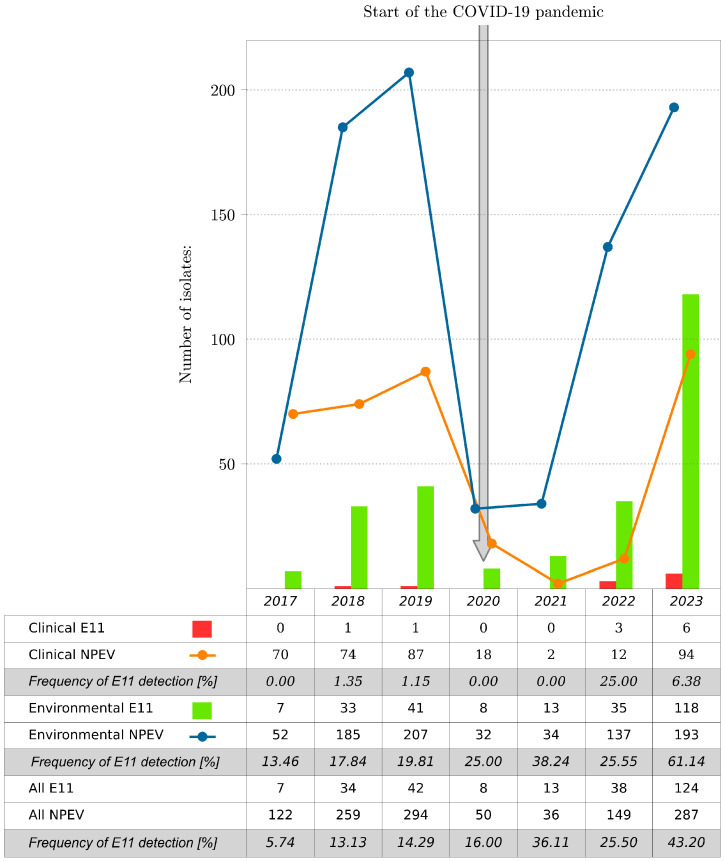
Number of E11 and all NPEV isolates by year, Poland, 2017–2023. The frequency of E11 detection is shown as the percentage of E11 to NPEV per year.

**Figure 2 viruses-16-01011-f002:**
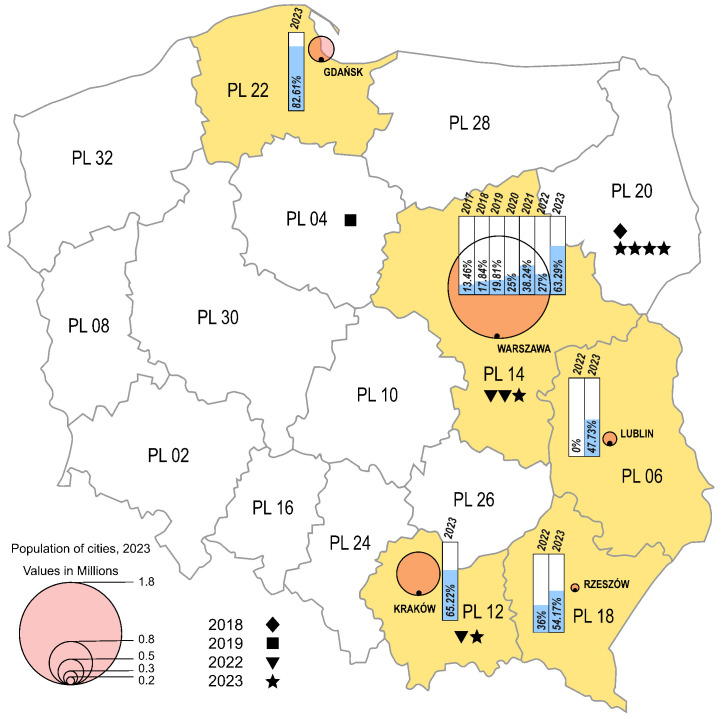
Frequency of E11 detection in Polish regions, 2017–2023. The voivodeships in which environmental surveillance was carried out are marked in yellow. Each city monitored by environmental surveillance is represented by a circle, with the size proportional to the population of the city (Warsaw—1.8 million; Krakow—0.8 million; Gdansk—0.5 million; Lublin—0.3 million; Rzeszów—0.2 million). Symbols (star, triangle, square, diamond) mark cases of E11 infections. The percentage of environmental E11-positive samples is shown in the bar charts.

**Figure 3 viruses-16-01011-f003:**
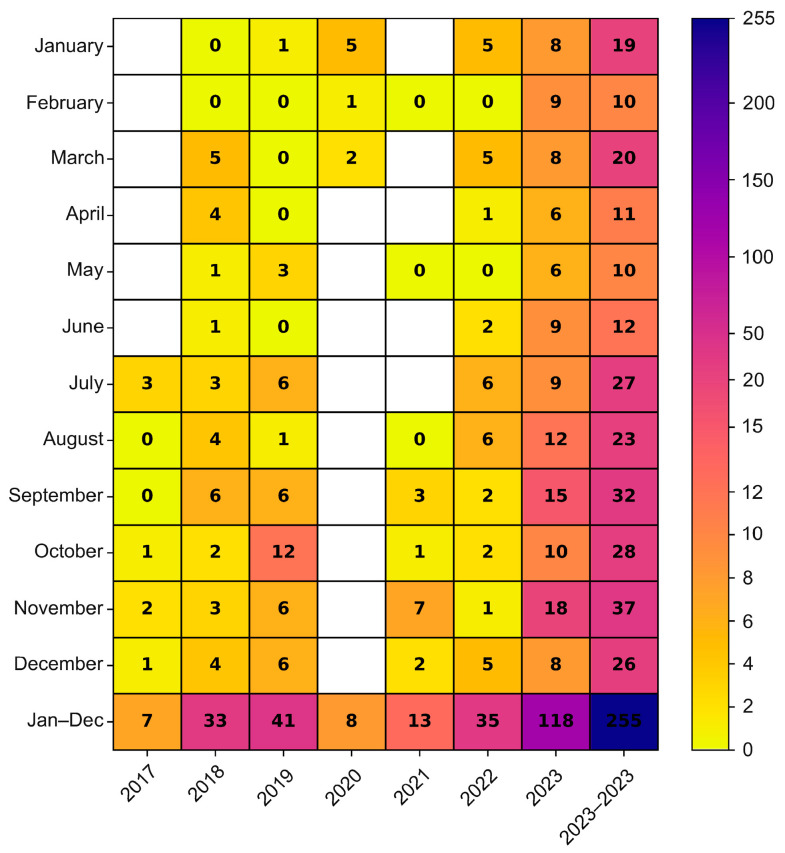
Monthly distribution of E11 isolated from environmental samples in Poland, 2017–2023. The number of environmental Polish E11 isolates detected by months was given in cells.

**Figure 4 viruses-16-01011-f004:**
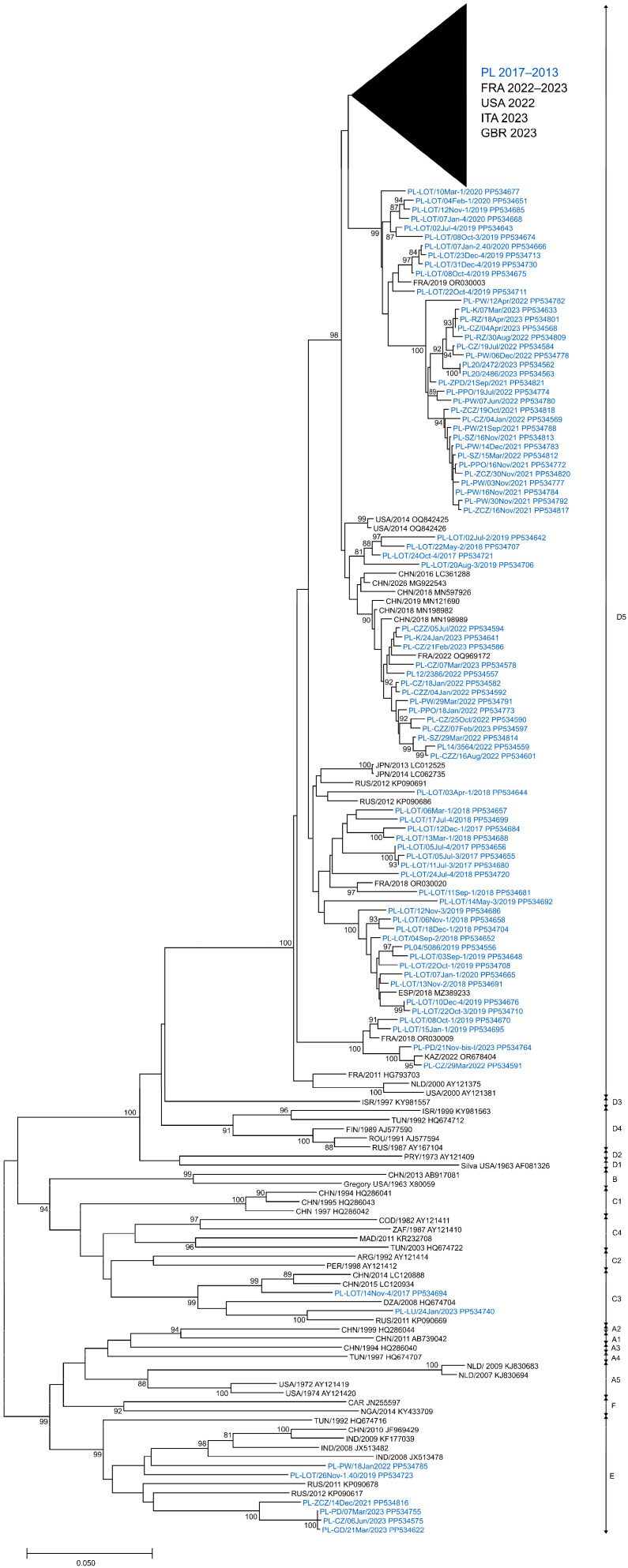
Phylogenetic tree of echovirus 11 complete 1D sequences from clinical and environmental samples in Poland (2017–2023) and other countries (1963–2023). The tree was constructed by the neighbor-joining method and evaluated with 1000 bootstrap pseudoreplicates. Only bootstrap values ≥ 80% are indicated. Genetic distances were calculated with the Kimura 2-parameter algorithm. Analyses were conducted in MEGA version X. Sequence identifiers consist of a country abbreviation, a year of detection and an accession number.

**Figure 5 viruses-16-01011-f005:**
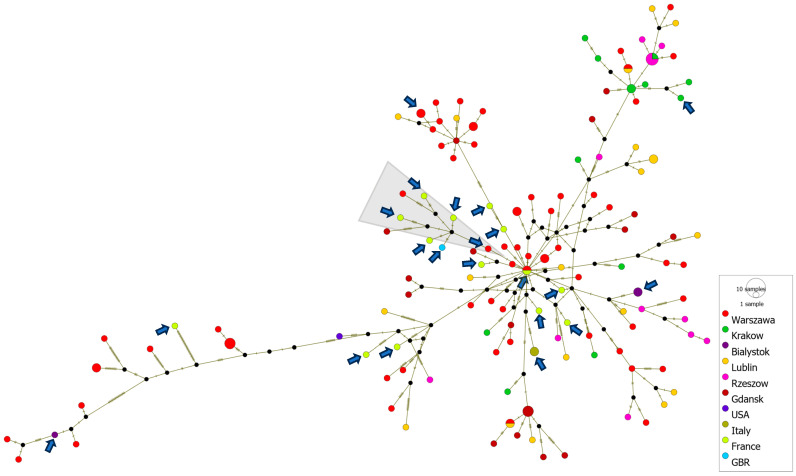
Median-joining haplotype network of 150 D5 genotype sequences constructed in PopArt. The analysis is based on entire VP1 gene sequences (876 nt) from clinical and environmental isolates collected in Poland from 2017 to 2023, as well as from other countries during 2022–2023. Each circle represents a unique haplotype, and its size reflects the number of strains expressing that haplotype. Crosshatches indicate the number of nucleotide differences between haplotypes. Color codes denote the geographic location of the strains. Clinical isolates are marked with arrows. The group of strains associated with severe E11 infections is marked with a gray triangle.

**Table 1 viruses-16-01011-t001:** Clinical characteristics of E11 cases, Poland, 2018–2023.

No.	Name of Strain Accession Number	Gender/ Age	Date of Onset	Case Type	Symptoms/Treatment/Response
1	PL20/4497/2018 PP534566	F/17 years	June 2018	mild	Meningitis, headaches, nausea, photophobia, fever, dehydration; CSF leukocyte count 7 cells/µL, CSF EV RNA: positive; Treatment: symptomatic; Response: recovered without sequelae;
2	PL04/5086/2019 PP534556	M/7 years	May 2019	ND	EV RNA in stool sample: positive;
3	PL12/2386/2022 PP534557	M/2 months	June 2022	mild	Fever, apathy, poor feeding, tachycardia, rash on the lower limbs; EV RNA in stool sample: positive;
4	PL14/3564/2022 PP534559	F/6 years	August 2022	mild	Meningitis; EV RNA in stool sample: positive;
5	PL14/4786/2022 PP534560	F/12 days	November 2022	severe	Severe hepatitis and multisystemic disease; EV RNA in stool sample: positive;
6	PL20/2472/2023 PP534562	F/11 days	January 2023	mild	Meningitis, irritability, fever, tachypnoea, tachycardia, elevated interleukin 6 (max. 102 pg/mL, low CRP, nasopharyngeal swab EV RNA: positive; CSF leukocyte count 8 cells/µL, CSF protein 77 mg/dL; CSF EV RNA: positive; Response: recovered without sequelae;
7	PL20/2486/2023 PP534563	F/13 days	January 2023	mild	Meningitis, fever, tachycardia, CSF leukocyte count 21 cells/µL; CSF EV RNA: positive; Treatment: symptomatic; Response: recovered without sequelae;
8	PL12/3636/2023 PP534558	M/6 days	September 2023	ND	CSF EV RNA: positive;
9	PL20/3983/2023 PP534564	M/2 years_6 months	October 2023	mild	Meningitis, vomiting, lethargy, poor feeding; CSF leukocyte count 15 cells/µL; CSF EV RNA: negative; EV RNA in stool sample: positive; Treatment: empirical antibiotics, symptomatic treatment; Response: recovered without sequelae;
10	PL20/3984/2023 PP534565	F/1 year_6 months	October 2023	mild	Meningitis, fever, headaches, vomiting, dehydration, preceded 2 weeks earlier by skin rash; CSF leucocyte count 263 cells/µL, CSF EV RNA: negative, EV RNA in stool sample: positive; Treatment: empirical antibiotics, symptomatic treatment; Response: recovered without sequelae;
11	PL14/4885/2023 PP534561	M/2 months	November 2023	ND	EV RNA in stool sample: positive;

Abbreviations: ND, no data; CSF, cerebrospinal fluid; EV, enterovirus; CRP, C-reactive protein.

**Table 2 viruses-16-01011-t002:** Nucleotide and amino acid divergence of E11 isolates detected in Poland (2017–2023), based on the full-length VP1 sequence analysis.

E11 Isolates	Year of Detection	No. of Isolates	Nucleotide Divergence [%]	Amino Acid Divergence [%]	Positive E11 Genotype Detection		
All	2017–2023	266	0.0–25.7	0.0–12.7	C3	D5	E
	2017	7	0.1–23.7	0.3–11.6	X	X	
	2018	34	0.0–9.3	0.0–2.7		X	
	2019	42	0.0–24.2	0.0–11.0		X	X
	2020	8	0.0–8.3	0.0–2.7		X	
	2021	13	0.1–24.7	0.0–10.6		X	X
	2022	38	0.0–24.4	0.0–11.0		X	X
	2023	124	0.0–25.3	0.0–12.3	X	X	X
Environmental	2017–2023	255	0.0–25.7	0.0–12.7	X	X	X
Clinical	2018–2019 2022–2023	11	0.0–8.5	0.0–2.7		X	
C3	2017, 2023	29	11.3	3.1			
D5	2017–2023	255	0.0–9.9	0.0–4.8			
E	2019, 2021–2023	9	0.0–15.1	0.0–3.8			

**Table 3 viruses-16-01011-t003:** Amino acid consensus of VP1 in Polish E11; strains of differential genotypes were compared with the prototype strain—Gregory.

VP1 Amino Acid Sites																																											
N terminus										B#		B-D loop				C-D loop		D#	D-E loop		E#	E-F loop		G-H loop		H#		H-I	I#			C terminus											
	7	9	19	23	43	45	48	54	63	78	80	84	86	87	89	96	109	117	131	132	144	157	161	186	215	219	221	227	238	245	247	267	268	270	271	273	276	279	280	283	289	290	292
Gregory	V	N	G	S	T	S	M	K	S	G	H	T	Q	T	L	S	I	V	S	R	I	V	A	I	H	I	V	S	M	V	A	T	P	N	V	D	T	N	Y	E	L	S	Y
C3	V	N	G	S/L	T	S	V	K	S	E	H	T/S	Q	T	L	S	I	V	T	Q	I	V	T	I	H	I	V	S	M	I	A	S	P	D	I	D	E/N	T	Y	D	V	S	H/Y
D5	I	S	S	T	V	G	M	R	T	E	Y	T	E	T	R/K	N	M	V	T	Q	V	T	A	V	N	L	M	P	V	V	A	S	S	N	I	D	D	T	Y	D	V	T	H
E	V	N	S	S	T	S	I	H	S	E	Y	E/T	T/A	S	L	N	M/L	I	T	Q	I	V	T	I	H	I	V	P	M	I/V	V	T	P	N	V/I	E	Q/N	N	Y/H	D	V	S	Y

## Data Availability

The raw data supporting the conclusions of this article will be made available by the authors on request.

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
