# Peer review of "Intensified Circulation of Echovirus 11 after the COVID-19 Pandemic in Poland: Detection of a Highly Pathogenic Virus Variant"

_viruses, 2024, doi:10.3390/v16071011_

Round 1

Reviewer 1 Report

Comments and Suggestions for Authors

Key results. The manuscript reports on the detection of echovirus 11 in Poland between 2017 and 2023, including the recent period during other countries reported the occurrence of severe infection cases in neonates. The study is timely and provides helpful and necessary (clinical and virological) data to understand the epidemiology of this significant enterovirus.

The approaches are valid; the data are of high quality. The topics addressed in this study would be critical and worthwhile. However, there are a number of issues that need consideration to improve the manuscript.

Minor comments

Materials & methods. The authors should provide the basic data to understand how the environmental surveillance is organised and was performed during the study period (methods, number of wastewater treatment plants, sampling periodicity...).

Figure 1. Could you indicate if there were any changes (increase or decrease) in sampling frequency in environmental surveillance during the study period. The reader would be would be interested to know the denominator, i.e., the total number of distinct samples tested per year.

Figure 2. What proportion of the national population is living in the voivodeships where the environmental surveillance is implemented?

Lines 162 - 163. I cannot understand what cytosis (lymphocytosis?) is exactly and the quantification unit of the biological parameter (check also Table 1 and explain “mcL” in a footnote). Please, clarify and check how this parameter is evaluated.

Table 1. the title should be revised because the first case is dated in 2018 not 2017.

Figure 4. The phylogenetic should be redrawn for more clarity. Sequence designations are unreadable. I can see that many sequences are strictly similar. The authors should select only one exemplar among redundant sequences, this would improve the overall tree readability. In addition, it would be more informative to the reader to collapse "old" clades and expand the most recent cluster (2017 - 2023). Alternatively, the authors can include an insert in the figure to show the recent cluster.

Line 220 (legend to Fig. 5). The authors should change "isolate" to "sequence".

Lines 244 to 308. The beginning of the discussion section is difficult to follow because comments on clinical surveillance alternate with comments on environmental surveillance in different paragraphs. The clarity of the discussion would benefit from separating these two aspects in two paragraphs.

Lines 324 - 326. I cannot understand the exact meaning of the sentence "Phylogenetic dates from...". Please, could you explain.

Line 355. Please, consider revise the sentence as suggested "the close genetic relationships between isolates recovered in Poland and other European countries"...

Comments on the Quality of English Language

Some sentences (not all reviewed in the comments) are not worded very well and could be improved.

Reviewer 2 Report

Comments and Suggestions for Authors

In this manuscript, Gad et al. describe a comprehensive study of echovirus 11 circulation before and after the COVID-19 pandemic in Poland, detecting the highly pathogenic variant of the virus that has been reported in Europe, Asia and the United States in recent years.  The authors obtained a very informative and important result based on their well design and complete study where wisely included the clinical and environmental surveillance, virus isolation, clinical, demographic and phylogenetic analysis. Undoubtedly could be considered for publication after so minor datils that in my consideration could be improved.

Minor Issues

1. Referring to the Silva strain as a prime instead of a prototype as described by Li et al. In my opinion, this comparison is not necessary, since the majority of studies, including the Fields Virology editions, refer to the Gregrory strain as the reference strain for Echo 11, but it may not be superfluous when considering possible new findings from a molecular point of view that could be related to the new genetic variant.

2. In the phylogenetic tree, for example, the Polish sequences could be highlighted with a different colour. The authors can also collapse the branches in which all the sequences are from the same year and countries to increase the size of the figure. In the legend, describe how each sequence is named (country, year, accession number).

3. In Table 3, I think it would be more informative if the Gregory strain was distinguished and placed at the top above the Polish, as well as the last row followed by the amino acid sites.

4. In the discussion, the authors could reorganise the ideas, grouping them by topic, first everything related to the distribution of this genetic variant, including the results of the phylogenetic analysis, which is the main objective of the study, trying to include everything related to the results of the study to what is reported in the world in a more organised and less repetitive way. Subsequently, everything related to environmental monitoring, which is the other main objective of the study, and to organise, if possible, the other studies chronologically and by country to avoid repeating ideas and studies from the same countries in different years.

5. Limitations may include the geographic representativeness of the central western part of the country and the use of the Hep2 cell line for viral isolation of ECHO 11 as described in other studies.

Comments on the Quality of English Language

Just a general review of how some ideas have been written. 
